# Electromyographic analysis of stomatognathic muscles in elderly after hippotherapy

**Edneia Corrêa de Mello**[1,2]*, **Simone Cecílio Hallak Regalo**[1], **Luanna Honorato Diniz**[2], **Janaine Brandão Lage**[2], **Mariane Fernandes Ribeiro**[2], **Domingos Emanuel Bevilacqua Junior**[2], **Rodrigo César Rosa**[3], **Alex Abadio Ferreira**[4], **Mara Lúcia Fonseca Ferraz**[2], **Vicente de Paula Antunes Teixeira**[2], **Ana Paula Espindula**[2,3]

1 Department of Basic and Oral Biology, School of Dentistry University of São Paulo, Ribeirão Preto, São Paulo, Brazil, 2 General Pathology Discipline, Federal University of Triângulo Mineiro, Uberaba, Minas Gerais, Brazil, 3 Human Anatomy Discipline, Federal University of Triângulo Mineiro, Uberaba, Minas Gerais, Brazil, 4 Association of Parents and Friends of the Exceptional, Uberaba, Minas Gerais, Brazil

* edneia.mello@usp.br

## Abstract

The aging process affects the entire human body, including the stomatognathic system, and can trigger not only occlusal but also postural imbalances involving other muscular chains. Hippotherapy has been used to promote cervical, postural, and balance control in individuals with diverse impairments. The present study used electromyography to evaluate the masseter and temporal muscles in an elderly population pre- and post-hippotherapy. Participants included 17 individuals, mean age 66,5±7 years. Electromyographic recording of the bilateral masseter and temporal muscles was performed during the postural resting condition and activities that involved the active participation of these muscles in different conditions. The practitioner performed no other type of activity or exercise during the intervention because the objective is to evaluate the effect of the three-dimensional movement provided by the horse. Raw electromyographic data were tabulated using commercially available software (IBM® SPSS® Statistics 234.0) and subjected to statistical analysis, in which $p \leq 0.05$ was considered to be statistically significant. Post-hippotherapy, there was lower electromyographic activity for the masseter and temporalis muscles in all the static mandibular tasks, with significant effect for time for the right temporal muscle (p = 0.038), the left temporal muscle (p = 0.028) and in the all dynamic mandibular tasks for the left temporal muscle (p = 0.025) and the left masseter muscle (p = 0.027). Hippotherapy promotes a reduction in the myoelectric activity of the masticatory muscles of elderly individuals.

## Introduction

The proper and harmonic functioning of the stomatognathic system plays a fundamental role in the well-being of humans. When physiological factors compromise this system, an imbalance occurs in masticatory muscular dynamics that can extend to other muscular chains, causing a postural imbalance in the distal regions of the body, such as changes in hip alignment [1]. This imbalance generates a predisposition to falls, representing the highest risk for

**Data Availability Statement:** All relevant data are within the manuscript.

**Funding:** The authors received no specific funding for this work.

**Competing interests:** NO authors have competing interests

morbidity and mortality in the elderly population [2, 3]. However, when malocclusions are corrected, there may be improvements in static and dynamic balance [4, 5]. It has been established that physical activity promotes a reduction in the risk for falls in this population, and is indicated as a non-drug treatment for several diseases that affect this population, including sarcopenia. Sarcopenia, age-dependent loss of muscle mass and function, is an increasingly prevalent and affects as much as 33% of the elderly population [6]. Quantitative and qualitative losses of skeletal muscle are associated with several adverse health outcomes [7]. Repercussing in the stomatognathic system is believed to be one of the main reasons for the decrease in masticatory capacity and is related to atrophy of the jaw-lifting muscles [8]. Accordingly, it is recommended that elderly individuals remain as active as their condition allows [9].

Hippotherapy is a treatment strategy that utilizes equine movement as part of a comprehensive program of intervention for the attainment of functional outcomes. It is not intended to replace conventional treatment(s) but complementary therapy [10, 11]. This therapeutic activity facilitates and requires the entire individual's participation, thus contributing to the improvement in trunk strength, balance control, and motor coordination [12]. It has been demonstrated that the recreational use of therapeutic riding can improve coordination, motor skills, posture, and head control [13]. It is an integrated intervention program to achieve functional results, contributing to the rehabilitation of cardiopulmonary, musculoskeletal, and neuromuscular dysfunctions [14, 15]. The indication for this therapy has grown considerably, including for the elderly population [16, 17]."

The use of electromyography (EMG) to evaluate whether this three-dimensional movement influences the recruitment of masticatory muscles is important because it reflects the increasing diversity of clinical indications for hippotherapy. This study aimed to evaluate the effect of hippotherapy on the stomatognathic system. We hypothesized that hippotherapy could alter the myoelectric activity of the masticatory muscles. To our knowledge, this is the first study to assess the influence of equine movement on masticatory muscles in the elderly.

## Materials and methods

The present investigation was designed as an observational, descriptive and quantitative study. The project was evaluated, and approved by the Research Ethics Committee (CEP) of the Federal University of Triângulo Mineiro–UFTM under the protocol number 690.039, on 13/02/2014, by the UFTM Ethical Committee on the Use of Animals under protocol no. 266/2017, on 21/07/2017 and by the Brazilian Registry of Clinical Trials (ReBEC) under protocol RBR-2kw6p9, with access via the following link (http://www.ensaiosclinicos.gov.br/rg/?q=RBR-2kw6p9). The evaluation methods and intervention protocols used in this study adhered to the norms of Resolution 466/12 of the National Health Council on Research Involving Human Beings and the Law 11.794/08 Decree 6.899/09 by the National Council for Animal Control and Experimentation (CONCEA). Participants in this study provided written informed consent by signing a Free and Informed Consent Form.

### Study sample

Data from 90 individuals, who attended the Unit of Attention to the Elderly (UAI) of Uberaba, Minas Gerais, Brazil, including age, sex, lifestyle, diseases, and medication use, were reviewed. Individuals of both genders, aged between 60 and 79 years, with an average body mass index (BMI) considered normal for this age group. Individuals with physical impairment (due to stroke or post-surgical sequelae), uncontrolled epilepsy and hypertension, acute heart disease, spinal instabilities, severe cervical spine disorders, shoulder or hip dislocations, scoliosis evolving by 30 degrees or more, decubitus ulcers in the pelvic or lower limb region, uncontrollable

fear of horses, severe cervical spine dislocations/deformities, shoulder or hip dislocations, or uncontrollable fear of horses, were excluded. Ultimately, 17 individuals participated in the study, of whom 14 were female, and 3 were male, average age: 66,5 ± 7 years, with an average body mass index (BMI) of 25.6 kg/m$^2$ ± 8, which is considered to be a normal BMI for this age group.

## Hippotherapy

Hippotherapy interventions were performed at the Dr. Guerra Equine Therapy Center of the Association of Parents and Friends of the Exceptional (APAE) in the city of Uberaba, Minas Gerais, Brazil, which serves the area and provides an access platform. The treatment was performed with the horse in step for 30 min. Two trained horses, 18 and 9 years of age, with a height of 1.60 m and 1.62 m, respectively, were used at random. The selected riding material was the Australian saddle with feet in the stirrups for the first 15 min, and with the feet out of the stirrups for the remaining 15 min. No other type of activity or exercise was performed by the practitioner during the session because the objective was to evaluate the effect of the three-dimensional movement provided by the horse [17, 18]. The visits were conducted only by examiners previously qualified by the National Association of Equine-Assisted Therapy [19].

## EMG activity

The evaluation of muscular activity was performed before and immediately after a single intervention of hippotherapy. For the EMG evaluation, the masseter and temporalis muscles were analyzed in the following clinical conditions: rest (10s), right laterality (10s), left laterality (10s), protrusion (10s) and dental clenching with Parafilm M® (Pechinery Plastic Packaging, Batavia, IL, USA), folded (18x17x4mm, 245 mg) positioned between the occlusal surfaces of the superior and inferior first molars, bilaterally (10s). The EMG signals were collected during clinical conditions and verified using the root-mean-square (RMS) [20]. For the dynamic evaluation of forces during mastication, the efficiency of the masticatory cycles was verified utilizing the integrated linear envelope EMG of the masseter and temporalis muscles (bilaterally). The values were registered in units of (microvolt/seconds). The EMG signals were collected during habitual mastication of hard (5g of shelled peanuts), soft foods (5g of raisins without seeds), and non-habitual chewing with Parafilm M®, for 10 seconds [21]. Non-habitual chewing is a movement with dynamic, short-excursion mouth-opening and hinge-type movement, reported to decrease the effects of changing the length and muscle tension [22]. Before the exam, the subjects were informed about the types of food to be chewed, emphasizing that it was not necessary to swallow them after the electromyographic exam They could be disposed of in an approved container.

An electromyograph (SAS1000V8, EMG do Brazil Ltd., São José dos Campos, SP, Brazil) equipped with a data acquisition, control, storage, processing and analysis system was used. Before placement of the electrodes, the skin was cleaned with alcohol to eliminate any grease residues or pollution that may be present. Differential active electrodes were positioned. The reference electrode was placed on the left wrist, on the styloid process of the ulna. To ensure the correct location of the masseter and temporal muscles, specific MVIC maneuvers were performed, accompanied by palpation [23]. To accurately reproduce the position of the electrodes, markings were made on the face using a dermatographic pencil. During electromyographic recording, the environment was kept quiet, and subjects were positioned sitting with the trunk and head erect in a comfortable chair, such that they kept their eyes on the horizon, the soles of the feet resting on the ground, and arms resting on the thighs. The necessary instructions and explanations were given, asking the individual to remain as calm as possible, and to breathe slowly and calmly.

EMG data were collected at the institution's dental office, with temperature control (20˚) and lighting (penumbra). When comparing EMG activity on the same muscles, same individual, and day, the EMG signal does not need to be normalized [24]. Also, information is lost when the data are normalized [25]. Therefore, this study used raw data from the EMG signal.

## Statistical analysis

Raw EMG data were tabulated and subjected to statistical analysis using commercially available software (IBM® SPSS® Statistics 234.0). Data were analyzed by descriptive statistics (average and standard error) and Repeated Measures ANOVA with Bonferroni's post hoc. The factor 1 time (pre and post) and factor 2 clinical condition (rest, right laterality, left laterality, protrusion and dental clenching with Parafilm) were used for static mandibular tasks. The factor 1 time (pre and post) and factor 2 habitual and non-habitual chewing condition were used for the dynamic tasks of the mandible. The level of statistical significance was established as $p \leq 0.05$.

## Results

Initially, a pilot study was performed with 7 elderly participants, who were not included in the final sample. Data from this preliminary study were used to calculate the sample size. A post hoc sample size calculation was conducted considering a level of α = 0.05, a power of 95% for the primary outcome electromyographic activity during the raisins chewing condition [average of the right masseter muscle, pre-hippotherapy = 2.20 (0.56) and post-hippotherapy = 1.50 (0.60)], and effect size of 1.206. The minimum sample size obtained was 16 participants. The sample size calculation was performed with the G*Power software v 3.0.10 (Franz Faul, Kiel University, Kiel, Germany).

Post-hippotherapy, there was lower electromyographic activity for the masseter and temporalis muscles in all the static mandibular tasks, with significant effect for time for the right temporal muscle (p = 0.038) and the left temporal muscle (p = 0.028), indicating that, regardless of the clinical condition, the means at the end of the intervention were lower when compared to the initial mean, Table 1.

Post-hippotherapy, there was lower electromyographic activity for the masseter and temporalis muscles in the all dynamic mandibular tasks, with significant effect for time for the left temporal muscle (p = 0.025) and the left masseter muscle (p = 0.027), indicating that, regardless of the clinical condition, the means at the end of the intervention were lower when compared to the initial mean, Table 2.

## Discussion

The results validate the hypothesis that hippotherapy alters the myoelectric activity of the masticatory muscles. In the present study, EMG analysis was used to evaluate the performance of the masticatory muscular activity in different clinical conditions and muscular alterations resulting from hippotherapy. EMG has been widely used to assess facial muscles' performance in the physiological process of mastication [26, 27].

In the resting condition, the mandible is involuntarily suspended by reciprocal coordination of the lifting and lowering muscles — a position considered to be neutral. The muscles exhibit neuromuscular tonus that is in a state of passive resistance to the stretching of the fibers. Consequently, stimuli arrive at their motor units (nerve and muscle cells) alternately to avoid fatigue. This occurs through unconscious and automatic myotatic reflexes, maintaining the mandible in an antigravity and naturally relaxed position [28, 29]. In other words, although the muscles are not inactive, electromyographic silence characterizes this position [30]. Other

**Table 1. Average, standard error (±), Fisher (F), degrees of freedom (df) and statistical significance (*p*<0.05) of effect of time, clinical condition and time versus clinical condition of raw electromyographic data (μV) of the right and left temporal (T) and masseter muscles (M), in the clinical conditions: rest (R), right laterality (RL), left laterality (LL), protrusion (P), dental clenching with Parafilm M® (DEP), for the pre- and post-hippotherapy intervention.**

| Muscle | | Clinical Condition | | | | | Effect of Time | Effect of Clinical Condition | Effect Time x Clinical Condition |
|---|---|---|---|---|---|---|---|---|---|
| | | R | RL | LL | P | DEP | F (df)—*p* | F (df)—*p* | F (df)—*p* |
| **Right** | Pre | 24.14 | 30.01 | 21.81 | 21.87 | 28.19 | F (1;6) = 5.12 | F (4;64) = 1.57 | F (4;26,3) = 0.33 |
| | | ±6.14 | ±5.48 | ±4.53 | ±3.81 | ±5.58 | | | |
| **T** | Post | 21.40 | 21.02 | 18.58 | 19.76 | 28.17 | **0.038*** | 0.194 | 0.678 |
| | | ±3.92 | ±3.97 | ±4.40 | ±4.92 | ±4.95 | | | |
| **Left** | Pre | 33.09 | 31.74 | 29.99 | 30.29 | 35.44 | F (1;16) = 5.97 | F (4;64) = 1.38 | F (4;64) = 1.97 |
| | | ±6.43 | ±6.04 | ±6.34 | ±6.56 | ±3.95 | | | |
| **T** | Post | 14.46 | 28.92 | 27.84 | 23.64 | 32.47 | **0.028*** | 0.252 | 0.110 |
| | | ±5.59 | ±4.62 | ±4.73 | ±3.51 | ±5.31 | | | |
| **Right** | Pre | 24.84 | 29.80 | 25.25 | 26.25 | 28.2 | F (1;16) = 0.533 | F (4;64) = 0.301 | F (4;64) = 0.021 |
| | | ±6.54 | ±5.80 | ±6.10 | ±6.75 | ±5.84 | | | |
| **M** | Post | 23.64± | 26.72 | 24.29 | 22.89 | 26.84 | 0.476 | 0.876 | 0.999 |
| | | 6.49 | ±5.75 | ±6.56 | ±4.45 | ±5.14 | | | |
| **Left** | Pre | 18.58 | 25.29 | 24.93 | 22.41 | 32.01 | F (1;16) = 1.12 | F (4;64) = 3.51 | F (2,63;42.03) = 0.200 |
| **M** | | ±3.42 | ±4.02 | ±3.65 | ±2.27 | ±3.94 | 0.306 | 0.079 | 0.837 |

studies have reported that, in the resting position, there is electrical activity in the skeletal striated muscle [31, 32]. In this study, there was electrical activity in the resting position, however, in post-treatment, EMG activity was lower, with statistically significant differences in the left temporal, and without significance for the temporal and right masseter muscles. The left masseter maintained the same value. Such findings infer that hippotherapy decreased myoelectric activity. On each step, the center of gravity of the practitioner is displaced from its midline,

**Table 2. Average, standard error (±), Fisher (F), degrees of freedom (df) and statistical significance (*p*<0.05) of effect of time, clinical condition and time versus clinical condition of raw electromyographic data (ƒenv -μV) of the right and left temporal (T) and masseter muscles (M), in the habitual chewing condition (raisins and peanuts) and non-habitual (Parafilm M®), for the pre and post-hippotherapy intervention.**

| Muscle | | Habitual and Non-Habitual Chewing Condition | | | Effect of Time | Effect of Clinical Condition | Effect Time x Clinical Condition |
|---|---|---|---|---|---|---|---|
| | | Raisins | Peanuts | Parafilm | F (df)—*p* | F (df)—*p* | F (df)—*p* |
| **Right** | Pre | 4.97 | 4.64 | 5.37 | F (1;16) = 1.123 | F (2;32) = 0.734 | F (2;32) = 0.060 |
| | | ±0.93 | ±0.65 | ± 0.92 | | | |
| **T** | Post | 4.14 | 3.74 | 5.01 | 0.305 | 0.488 | 0.942 |
| | | ± 0.70 | ±0.89 | ±0.98 | | | |
| **Left** | Pre | 6.61 | 6.90 | 9.75 | F (1;16) = 6.134 | F (2;32) = 1.044 | F (2;32) = 0.965 |
| | | ±1.04 | ±1.21 | ±2.58 | | | |
| **T** | Post | 5.52 | 6.59 | 5.79 | **0.025*** | 0.364 | 0.392 |
| | | ±1.03 | ±1.02 | ±0.93 | | | |
| **Right** | Pre | 10.70 | 9.14 | 10.81 | F (1;16) = 1.906 | F (2;32) = 0.303 | F (2;32) = 0.270 |
| | | ±2.36 | ±1.21 | ±1.25 | | | |
| **M** | Post | 8.38 | 8.72 | 9.14 | 0.186 | 0.741 | 0.765 |
| | | ±1.29 | ±1.13 | ±0.86 | | | |
| **Left** | Pre | 6.65 | 14.51 | 6.86 | F (1;16) = 5.886 | F (1,39;22.50) = 2.587 | F (1.22;19.52) = 1.765 |
| | | ±1.73 | ±4.78 | ±1.39 | | | |
| **M** | Post | 5.33 | 5.74 | 6.12 | **0.027*** | 0,112 | 0,201 |
| | | ±1.25 | ±1.03 | ±1.38 | | | |

causing an imbalance and then a rebalancing, providing restoration of the center of gravity in the support base. In this manner, the vestibular system is continuously prompted. Slow vestibular stimulation benefits the balance and relaxation of muscle tone throughout the body [33]. Equine-assisted therapy has been used to control anxiety, stress, and trauma [34, 35]. Stress, anxiety, tension and nervousness exert high impact on the masticatory muscles, causing an increase in myoelectric activity patterns [36, 37]. This situation occurs due to the unification of the system responsible for the emotions and social behaviors of the individual with the motor part of the central nervous system, which elevates myoelectric activity in the masticatory muscles [38].

During left and right lateral movements, there is a neuroanatomical muscular activation pattern with greater electromyographic activity in the temporal muscle of the ipsilateral side of the mandible (i.e., working side), while in the masseter muscle, there is more activity on the contralateral side (balance side or "I do not work") [39]. The results of the present study demonstrate this pattern of activation for the two time points analyzed. In the right laterality, in the post-treatment, EMG activity was lower in both masseters and the right temporal muscles. In the left temporal masseter, it was similar. In the left laterality, EMG activity was lower in the right temporal, similar in the right and left temporal masseter and more prominent in the left masseter muscles, without statistically significant differences.

During the protrusion condition, the pattern of muscular behavior required to maintain the position is revealed by the greater activation of the masseter muscles compared with the temporal muscles [40]. Data from this research corroborate this pattern. In post-treatment, the activity of the left masseter muscle was lower, but was similar in the other muscles (i.e., without statistically significant differences).

Under conditions of dental clenching in individuals who do not present muscular morpho-functional changes, myoelectric activity of the masseter muscle is higher than in the temporal muscle [41]. This is due to the morphological and functional characteristics of these muscles. Mandibular movements have a very short trajectory, requiring high speed and greater precision. The masseter is a powerful muscle with a force function that carries and supports the bones, protects and drives the power of movement, and elevates the jaw during various oral functions. While the temporal muscle has a purpose more related to velocity, being the first to be contracted in mandibular closure, it is considered to be a positioner of the mandible, because it better adjusts the direction of the movement, acting as synchronizer of movements [42, 43]. These findings reflect the results of this research because the EMG activity of the masseter muscles was more significant than the temporal muscles. In post-treatment, EMG activity of the masseters reached closer values, inferring more balanced muscle recruitment between the two muscles, which is positive given that unilateral vicious chewing is sufficient to cause asymmetry of the face [44].

After placing food in the mouth, the depressing and lifting muscles of the jaw collaborate significantly with the dynamics of chewing, composed of movements of isotonic contractions interspersed with periods of isotonic contraction [40]. An essential tool used for the analysis of masticatory cycles is the integral of the envelope of the EMG signal that analyzes only the periods of isometric contractions and signals possible changes in masticatory efficiency [45].

During habitual and non-habitual chewing, the myoelectric activity of the masseter muscle is higher than the temporal muscle because it exhibits more significant action potential [41, 45]. The chewing of peanuts, raisins, and Parafilm M results confirm the pattern of superior EMG activity of the masseters compared with the temporal muscles at both time points [42]. Post-treatment EMG activity was statistically lower for the right and left masseter muscles in the mastication of raisins. This is desirable, given that in healthy individuals, there is less recruitment of muscle fibers to perform the same masticatory function compared with

individuals with morpho-functional changes that generate stress and fatigue [45]. The EMG activity in the chewing of peanut and raisins were also lower but without statistically significant difference. Therefore, we recommend that future studies analyze the stomatognathic system in elderly individuals after treatment with more extended therapy.

## Conclusion

These findings suggest that hippotherapy promotes a reduction in the myoelectric activity of the masticatory muscles of the elderly.

## Author Contributions

**Conceptualization:** Vicente de Paula Antunes Teixeira, Ana Paula Espindula.

**Data curation:** Edneia Corrêa de Mello.

**Formal analysis:** Janaine Brandão Lage, Domingos Emanuel Bevilacqua Junior.

**Investigation:** Edneia Corrêa de Mello, Luanna Honorato Diniz.

**Software:** Rodrigo César Rosa.

**Supervision:** Ana Paula Espindula.

**Writing – original draft:** Alex Abadio Ferreira, Mara Lúcia Fonseca Ferraz.

**Writing – review & editing:** Simone Cecílio Hallak Regalo, Mariane Fernandes Ribeiro.

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
