## [Decision Letter · Decision Letter 0]

8 May 2020

PONE-D-20-07826

Analysis of the stomatognathic system in elderly individuals after hippotherapy

PLOS ONE

Dear Mrs de Mello,

Thank you for submitting your manuscript to PLOS ONE. After careful consideration, we feel that it has merit but does not fully meet PLOS ONE’s publication criteria as it currently stands. Therefore, we invite you to submit a revised version of the manuscript that addresses the points raised during the review process.

Please respond to the comments from each Reviewer. I agree with Reviewer 2 that a change in title is appropriate. A better title might be, "Electromyographic analysis of stomatognathic muscles in elderly individuals after hippotherapy".

Separately, I feel it is imperative that you address, in your revised manuscript, some issues relating to your inferential statistics. In particular, please address the issue of multiple comparisons. In analyzing your results, you conducted many t-tests; at least 20. Did you make any correction for multiple comparisons, for example, in your criterion alpha? The submitted manuscript appears to suggest that you did not make such corrections. With appropriate correction for multiple comparisons, it is not clear that any of your effects would meet the criteria for statistical significance. 

We would appreciate receiving your revised manuscript by Jun 21 2020 11:59PM. To enhance the reproducibility of your results, we recommend that if applicable you deposit your laboratory protocols in protocols.io, where a protocol can be assigned its own identifier (DOI) such that it can be cited independently in the future. For instructions see: http://journals.plos.org/plosone/s/submission-guidelines#loc-laboratory-protocols

We look forward to receiving your revised manuscript.

Kind regards,

Thomas A Stoffregen, PhD

Academic Editor

PLOS ONE

Journal Requirements:

2. Please include your tables as part of your main manuscript and remove the individual files. Please note that supplementary tables should be uploaded as separate "supporting information" files.

"The authors are grateful for the financial support received for this research, provided by the following institutions: National Council for Scientific and Technological Development (CNPq), Coordination of Improvement of Higher-Level Personnel (CAPES), Foundation for Research Support of the State of Minas Gerais (FAPEMIG), Uberaba Teaching and Research Foundation (FUNEPU) and Association of Parents and Friends of the Exceptional of Uberaba (APAE)."

"No"

Reviewers' comments:

Reviewer's Responses to Questions

**Comments to the Author**

1. Is the manuscript technically sound, and do the data support the conclusions?

Reviewer #1: Yes

Reviewer #2: Yes

2. Has the statistical analysis been performed appropriately and rigorously? 

Reviewer #1: Yes

Reviewer #2: Yes

3. Have the authors made all data underlying the findings in their manuscript fully available?

Reviewer #1: Yes

Reviewer #2: Yes

4. Is the manuscript presented in an intelligible fashion and written in standard English?

Reviewer #1: Yes

Reviewer #2: Yes

5. Review Comments to the Author

Reviewer #1: Manuscript ID: PONE-D-20-07826

“Analysis of the stomatognathic system in elderly individuals after hippotherapy”

Level of interest: I have revised this manuscript, this interesting article should be accepted for publication.

Quality of written English: Well done

Statistical review: Right

Declaration of competing interests: I have no competing interests to declare, hold no shares.

“Analysis of the stomatognathic system in elderly individuals after hippotherapy”

The authors state that the aim of the submitted manuscript was to identify how hippotherapy may alter the myoelectric activity of the masticatory muscles in elderly participants. This is an interesting and worthwhile observational, descriptive and quantitative research. Authors have put in a good effort on the project. Moreover, the research in my opinion is well written.

There are several details that could be improved on this manuscript.

Overall organization of the manuscript is good but further improvements in this area will significantly improve clarity of the paper. It is striking that the Method section is on page 11. The results are found on page 4 of this paper. Does not agree with the guidelines of the magazine. Please check it. Thank you

In addition to this, I include minor reorganization of a few parts of the manuscript, and significant rewriting of certain sections as suggested below:

1) INTRODUCTION: Line 68 to 71 on page 3: Please could you add the following reference in these lines?

Martín-Valero, R., Vega-Ballón, J., & Perez-Cabezas, V. (2018). Benefits of hippotherapy in children with cerebral palsy: A narrative review. European Journal of Paediatric Neurology, 22(6), 1150–1160. http://doi.org/10.1016/j.ejpn.2018.07.002

It is necessary to highlight the previously published results that found an increase in trunk strength and balance control. Hippotherapy is a treatment strategy that utilises equine movement as part of a comprehensive programme of intervention for the attainment of functional outcomes. It has been demonstrated tha the recreational use of therapeutic riding can improve coordination, motor skills, posture and control of the head.

2) Line 74 to 79 on Page 4. Please, I would recommend you that you can rewrtite these phrases.

MATERIAL AND METHODS:

3) Is the study registered in clinical trials with a reference number? Please, would you mind adding the reference number, Thank you in advance.

4) Line 120 to 125 on page 5 and 6 : Please could you add any reference in these lines?

5) Rephrasing is needed in the following area to enhance clarity:

Line 127 to 141 on page 6, please, could you improve the wording in these lines? Please could you add any reference in these lines?

6) DISCUSSION:

Line 235 to 242 on Page 10. Please, I would recommend you that you can rewrtite these phrases.

7) CONCLUSION:

Line 260 to 262 on Page 11. Please, I would recommend you that you can rewrtite these phrases.

Reviewer #2: Dear authors,

This is an interesting article related the use of hyppotherapy and EMG for the sthomarognathic system.

I find the introduction section adequate and the aim of the study is correctly reported.

Material and methods section explain clearly the study protocol used

Results section shows the data obtained in the study and they are correctly explained in the discussion section.

References appears adequate.

I would like to suggest a modification in the title, in order that the readers can understand that this study is made with EMG.

6. PLOS authors have the option to publish the peer review history of their article (what does this mean?). If published, this will include your full peer review and any attached files.

Reviewer #1: No

Reviewer #2: No

---

## [Author Response · Author response to Decision Letter 0]

19 Jun 2020

Dear Professor Thomas A Stoffregen, 

We were pleased to have an opportunity to revise our manuscript entitled "Electromyographic analysis of stomatognathic muscles in elderly individuals after hippotherapy." In the revised manuscript, we have carefully considered the editors' and reviewers' suggestions, and we adjusted some paragraphs accordingly. Below are the points we address and respond to each point raised by the editor and the reviewer. The responses to the editor's and reviewer's comments are below and are color-coded as follows: a) Comments from editors or reviewers are colored in red b) Our responses are shown under each comment as standard text. Overall, the comments were constructive, and we are appreciative of such constructive feedback on our original submission. After addressing the issues raised, we feel the quality of the paper is much improved. 

Best regards, 

Edneia Corrêa de Mello

 

Responses to editors’ comments: 

In analyzing your results, you conducted many t-tests; at least 20. Did you make any correction for multiple comparisons, for example, in your criterion alpha? The submitted manuscript appears to suggest that you did not make such corrections. With appropriate correction for multiple comparisons, it is not clear that any of your effects would meet the criteria for statistical significance.

To avoid the type I error, common when many t-tests are conducted; we opted for Repeated Measured ANOVA with Bonferroni’s post hoc. We also chose to work with the electromyography data because much information is lost when normalizing the data (lines 152 - 161, on page 7). 

On the title page, we added the department and listed the corresponding author's initials in parentheses after the email address. We adjusted the names of the files. We hope that it now fits the style requirements, as described in the referred templates. 

2. Please include your tables as part of your main manuscript and remove the individual files. Please note that supplementary tables should be uploaded as separate "supporting information" files.

We included our tables as part of the main manuscript and removed the individual files.

"The authors are grateful for the financial support received for this research, provided by the following institutions: National Council for Scientific and Technological Development (CNPq), Coordination of Improvement of Higher-Level Personnel (CAPES), Foundation for Research Support of the State of Minas Gerais (FAPEMIG), Uberaba Teaching and Research Foundation (FUNEPU) and Association of Parents and Friends of the Exceptional of Uberaba (APAE)."

"No"

a. Please clarify the sources of funding (financial or material support) for your study. List the grants or organizations that supported your study, including funding received from your institution.

d. If you did not receive any funding for this study, please state: “The authors received no specific funding for this work.”

We removed all funding-related text from the manuscript and would like to keep our Funding Statement as we wrote in our Cover Letter.

 

Responses to reviewer #1:

There are several details that could be improved on this manuscript.

Overall organization of the manuscript is good but further improvements in this area will significantly improve clarity of the paper. It is striking that the Method section is on page 11. The results are found on page 4 of this paper. Does not agree with the guidelines of the magazine. Please check it. Thank you.

The Method section on page 11 was removed from this paper. “Non-habitual chewing is a standardized short-opening, flap-type, mouth-to-mouth movement required to reduce the effects of changing length and muscle tension typical of dynamic records. Thus, it eliminates some interference factors that act in chewing, such as swallowing between masticatory cycles, chewing frequency, food texture, and masticatory preference.”

The results found on page 4 were transferred to the Results section (lines 164 - 171, on page 6 and 7). 

In addition to this, I include minor reorganization of a few parts of the manuscript, and significant rewriting of certain sections as suggested below: 

1) INTRODUCTION: Line 68 to 71 on page 3: Please could you add the following reference in these lines? 

Martín-Valero, R., Vega-Ballón, J., & Perez-Cabezas, V. (2018). Benefits of hippotherapy in children with cerebral palsy: A narrative review. European Journal of Paediatric Neurology, 22(6), 1150–1160. http://doi.org/10.1016/j.ejpn.2018.07.002

It is necessary to highlight the previously published results that found an increase in trunk strength and balance control. Hippotherapy is a treatment strategy that utilizes equine movement as part of a comprehensive program of intervention for the attainment of functional outcomes. It has been demonstrated that the recreational use of therapeutic riding can improve coordination, motor skills, posture and control of the head.

Thank you for the indication, we rewrote this paragraph and added the reference [13] (lines 61 – 71, on page 3). 

2) Line 74 to 79 on Page 4. Please, I would recommend you that you can rewrite these phrases.

We rewrote these phrases (lines 74 – 78, on page 4).

MATERIAL AND METHODS:

3) Is the study registered in clinical trials with a reference number? Please, would you mind adding the reference number, Thank you in advance.

The study registered in clinical trials with this reference number RBR-2kw6p9, and access is found on this link (http://www.ensaiosclinicos.gov.br/rg/?q=RBR-2kw6p9) (lines 86 and 87, on page 4).

4) Line 120 to 125 on page 5 and 6: Please could you add any reference in these lines?

We added references on these lines [17,18] [19] (lines 116 and 117, on page 5).

5) Rephrasing is needed in the following area to enhance clarity:

Line 127 to 141 on page 6, please, could you improve the wording in these lines? Please could you add any reference in these lines?

We rewrote this paragraph and added references [21] [22] (lines 119 - 136, on page 6 and 7).

6) DISCUSSION:

Line 235 to 242 on Page 10. Please, I would recommend you that you can rewrite these phrases.

We rewrote this paragraph (lines 250 - 255, on page 12).

7) CONCLUSION:

Line 260 to 262 on Page 11. Please, I would recommend you that you can rewrite these phrases.

We rewrote this paragraph (lines 270 and 271, on page 13).

 

Responses to reviewer #2:

I would like to suggest a modification in the title, in order that the readers can understand that this study is made with EMG.

We modified the title: Electromyographic analysis of stomatognathic muscles in elderly after hippotherapy 

Thank you again for your time and effort, 

Best regards

Edneia Mello

---

## [Editor Report · Decision Letter 1]

24 Jun 2020

PONE-D-20-07826R1

Electromyographic analysis of stomatognathic muscles in elderly after hippotherapy

PLOS ONE

Dear Dr. de Mello,

Thank you for submitting your manuscript to PLOS ONE. After careful consideration, we feel that it has merit but does not fully meet PLOS ONE’s publication criteria as it currently stands. Therefore, we invite you to submit a revised version of the manuscript that addresses the points raised during the review process.

I have noted elsewhere details about requested revisions.

We look forward to receiving your revised manuscript.

Kind regards,

Thomas A Stoffregen, PhD

Academic Editor

PLOS ONE

Additional Editor Comments (if provided):

Thank you for your careful revisions. Please note additional requests from Reviewer 1. As Editor, I continue to feel the need for more information about your statistical analysis. In the Section on Statistical Analysis (Line 156 FF), please state explicitly the factors in your ANOVAs. In addition, please revise Tables 1 and 2 to include the F values and degrees of freedom associated with each p-value. Once these matters are cleared up, your manuscript should be ready for publication.

---

## [Author Response · Author response to Decision Letter 1]

5 Aug 2020

We added more information about our statistical analysis. We explain the factors of the ANOVAs In the Section on Statistical Analysis. We have also included the F values and degrees of freedom associated with each p-value in Tables 1 and 2.

---

## [Editor Report · Decision Letter 2]

10 Aug 2020

Electromyographic analysis of stomatognathic muscles in elderly after hippotherapy

PONE-D-20-07826R2

Dear Dr. de Mello,

We’re pleased to inform you that your manuscript has been judged scientifically suitable for publication and will be formally accepted for publication once it meets all outstanding technical requirements.

Kind regards,

Thomas A Stoffregen, PhD

Academic Editor

PLOS ONE
---

## [Editor Report · Acceptance letter]

12 Aug 2020

PONE-D-20-07826R2 

Electromyographic analysis of stomatognathic muscles in elderly after hippotherapy 

Dear Dr. de Mello:

I'm pleased to inform you that your manuscript has been deemed suitable for publication in PLOS ONE. Congratulations! Your manuscript is now with our production department. 

Kind regards, 

on behalf of

Dr. Thomas A Stoffregen 

Academic Editor

PLOS ONE